# Characterization of the Peroxisomal Proteome and Redox Balance in Human Prostate Cancer Cell Lines

**DOI:** 10.3390/antiox13111340

**Published:** 2024-11-01

**Authors:** Mohamed A. F. Hussein, Celien Lismont, Cláudio F. Costa, Hongli Li, Frank Claessens, Marc Fransen

**Affiliations:** 1Laboratory of Peroxisome Biology and Intracellular Communication, Department of Cellular and Molecular Medicine, KU Leuven, 3000 Leuven, Belgium; mohamed.hussein@kuleuven.be (M.A.F.H.); celien.lismont@kuleuven.be (C.L.); claudiofcosta@live.com.pt (C.F.C.); hongli.li@kuleuven.be (H.L.); 2Department of Biochemistry, Faculty of Pharmacy, Assiut University, Asyut 71515, Egypt; 3Laboratory of Molecular Endocrinology, Department of Cellular and Molecular Medicine, KU Leuven, 3000 Leuven, Belgium; frank.claessens@kuleuven.be

**Keywords:** peroxisomes, prostate cancer, catalase, hydrogen peroxide, androgen receptor, R1881

## Abstract

Prostate cancer (PCa) is associated with disruptions in cellular redox balance. Given the intricate role of peroxisomes in redox metabolism, we conducted comprehensive proteomics analyses to compare peroxisomal and redox protein profiles between benign (RWPE-1) and malignant (22Rv1, LNCaP, and PC3) prostate cell lines. Our analyses revealed significant enrichment of the “peroxisome” pathway among proteins notably upregulated in androgen receptor (AR)-positive cell lines. In addition, catalase (CAT) activity was consistently higher in these malignant cell lines compared to RWPE-1, which contrasts with previous studies reporting lower CAT levels and increased H_2_O_2_ levels in PCa tissues compared to adjacent normal tissues. To mimic this clinical scenario, we used RNA interference to knock down CAT expression. Our results show that reduced CAT levels enhanced 22Rv1 and LNCaP cell proliferation. R1881-induced activation of AR, a key driver of PCa, increased expression of the H_2_O_2_-producing peroxisomal β-oxidation enzymes acyl-coenzyme A oxidase 1 and 3, reduced CAT expression and activity, and elevated peroxisomal H_2_O_2_ levels. Considering these changes and other antioxidant enzyme profile alterations, we propose that enhanced AR activity in PCa reduces CAT function, leading to increased peroxisomal H_2_O_2_ levels that trigger adaptive stress responses to promote cell survival, growth, and proliferation.

## 1. Introduction

Prostate cancer (PCa) ranks as the second most prevalent cancer and the fifth leading cause of cancer-related mortality among men worldwide. According to the latest estimates from the Global Cancer Observatory, there were 1,467,854 new cases and 397,430 deaths from PCa in 2022 [1]. The androgen receptor (AR), a nuclear hormone receptor essential for the normal development and function of the prostate gland, is the main driver of PCa. As a ligand-dependent transcription factor, the AR translocates from the cytosol to the nucleus upon binding to its cognate ligand. In the nucleus, it binds to DNA hormone response elements, regulating the transcription of its target genes [2]. Androgen deprivation therapy stands as the main treatment for locally advanced and metastatic PCa. However, most cases eventually develop resistance to therapy, progressing to a more aggressive stage known as castration-resistant PCa [3].

Tumorigenesis is consistently linked with abnormal cellular redox homeostasis [4]. Excessive production of reactive oxygen species (ROS) is associated with the uncontrolled activation of oncogenes and the enhancement of growth signaling pathways in cancer [4]. In PCa, disruptions in ROS signaling are linked to disease development and progression to more aggressive forms [5,6]. For example, dysregulation of redox-related signaling pathways (e.g., downregulation of nuclear factor erythroid 2-related factor 2 (Nrf-2) [7]), antioxidant enzymes (e.g., loss of glutathione S-transferase P (GSTP1) [6,8]), and elevated hydrogen peroxide (H_2_O_2_) levels [9,10] have all been observed. However, it remains unclear whether these disturbances in ROS homeostasis drive PCa development or are merely a consequence of it. In addition, the impact of disruptions in AR signaling on redox balance and how these changes affect therapeutic responses are still uncertain.

Peroxisomes are membrane-bound, multifunctional organelles involved in various key metabolic processes, including (i) β-oxidation of very long-chain fatty acids, (ii) α-oxidation of 3-methyl-branched chain fatty acids (e.g., phytanic acid), (iii) biosynthesis of ether lipids (e.g., plasmalogens), (iv) degradation of ROS, such as H_2_O_2_, (v) synthesis of polyunsaturated fatty acids (e.g., docosahexaenoic acid), (vi) glyoxylate detoxification, and (vii) biosynthesis of primary bile acids [11]. Increasing evidence suggests that peroxisomes are central hubs for cellular redox regulation [12]. These organelles contain numerous enzymes that generate H_2_O_2_ (e.g., acyl-CoA oxidases, D-amino acid oxidases, L-α-hydroxy acid oxidases, polyamine oxidases, and L-pipecolic acid oxidases) as well as others (e.g., catalase, glutathione S-transferase kappa 1, peroxiredoxin 1 and 5, and mitochondrial amidoxime reducing component 2) that directly or indirectly detoxify this ROS species [13,14,15]. In certain tissues, such as the liver, peroxisomes contribute up to 35% of the total H_2_O_2_ produced [16].

Catalase (CAT) is a tetrameric heme-containing peroxidase capable of reducing H_2_O_2_ into water and oxygen. Despite its low affinity for H_2_O_2_ (apparent Km: 80 mM), CAT exhibits a remarkably high turnover rate (apparent Vmax: 587,000 μmol H_2_O_2_/μmol heme/second) [17]. Consequently, it is regarded as an efficient H_2_O_2_ scavenger, particularly in environments with higher H_2_O_2_ concentrations. CAT is the most abundant peroxisomal matrix protein, targeted to peroxisomes via its non-canonical peroxisomal targeting sequence (PTS)-1 (-KANL), which is recognized by PEX5, the cycling PTS-1 import receptor. Due to (i) the relatively weak nature of this targeting sequence compared to the canonical -SKL, and (ii) PEX5’s role as a redox stress sensor, a portion of CAT often remains in the cytosol [18].

H_2_O_2_, once considered just a toxic byproduct of metabolic reactions, is now recognized as a crucial cellular signaling molecule. At physiological levels, it serves as a primary messenger, introducing post-translational oxidative modifications into specific protein targets involved in metabolic and stress reactions, such as AMP-activated protein kinase, Forkhead box protein O, mammalian target of rapamycin, hypoxia-inducible factor 1α, and cellular tumor antigen p53 [19]. These modifications can control downstream physiological functions, including growth, development, and steroidogenesis. However, when H_2_O_2_ levels exceed physiological thresholds, they disrupt the redox balance and cause a wide range of deleterious effects, such as DNA damage, growth arrest, and cell death [19].

The connection between peroxisomes and PCa was first identified over 20 years ago. Early studies using immunohistochemical analysis consistently found lower CAT expression in prostate adenocarcinoma compared to adjacent non-malignant tissue [20,21,22]. Meta-analyses later suggested that the C262T polymorphism in CAT, associated with reduced enzymatic activity, might be a risk factor for PCa [23,24,25]. In addition, various reports have observed changes in the expression levels of specific peroxisome-associated enzymes [26,27,28,29], including the upregulation of α-methylacyl-CoA racemase (AMACR), acyl-coenzyme A oxidase 3 (ACOX3), 2,4-dienoyl-CoA reductase 2 (DECR), and hydroxysteroid 17 beta dehydrogenase 4 (HSD17B4). Furthermore, it has been demonstrated that the peroxisomal branched-chain fatty acid oxidation pathway is upregulated in PCa [29] and that the peroxisomal membrane protein PMP34 [30] and the peroxisomal β-oxidation enzyme DECR2 [28] contribute to treatment resistance in advanced PCa. Despite these findings, many uncertainties remain regarding the role of peroxisomes in PCa progression and their overall impact on the disease.

In this report, we focus on the potential role of peroxisomes in regulating redox homeostasis in PCa cells. Our goals were to (i) examine variations in the peroxisomal redox state across various PCa cell lines, (ii) document changes in the peroxisomal and redox-related proteome between benign and malignant prostate cell lines in an unbiased manner, (iii) assess how changes in CAT activity impact cell growth behavior, and (iv) explore how AR signaling influences the subcellular redox environment and the expression profiles of peroxisomal and redox proteins. In summary, our findings highlight the importance of peroxisomal H_2_O_2_ metabolism in PCa biology and support the potential of CAT as a promising therapeutic target.

## 2. Materials and Methods

### 2.1. Cell Lines, Tissue Culture, and Treatment

Four prostate cell lines were used in this study: (i) RWPE-1, a human prostate epithelial cell line that has been immortalized using human papillomavirus 18 (CRL-3607, ATCC, Manassas, VA, USA); (ii) 22Rv1, a malignant prostate cell line derived from a localized tumor (CRL-2505, ATCC; (iii) LNCaP, a metastatic (lymph nodes) PCa cell line (CRL-1740, ATCC; and (iv) PC3, another metastatic (bone) PCa cell line (CRL-1435, ATCC). The cells were cultured in Eagle’s regular minimum essential medium (MEM) (M2279, Sigma-Aldrich, St.-Louis, MO, USA) supplemented with 10% (*v*/*v*) fetal bovine serum (FBS, S181B, Avantor, Leuven, Belgium), 2 mM ultraglutamine-1 (BE17-605E/U1, Lonza, Verviers, Belgium), and 0.2% (*v*/*v*) Mycozap (VZA-2012, Lonza) at 37 °C in a humidified 5% CO_2_ incubator. In the case of RWPE-1, the cells were also cultured in a keratinocyte serum-free medium (KSFM) (131-500A, a proprietary formulation from Sigma-Aldrich).

In experiments involving LNCaP cells treated with the AR agonist R1881 (E3164-000, Steraloids, Newport, RI, USA), the cells were cultured in phenol red-free MEM (51200046, Gibco, Billings, MT, USA) supplemented with 10% charcoal-stripped FBS (S181F-100, Avantor), 2 mM ultraglutamine-1, and 0.2% (*v*/*v*) Mycozap. R1881 was dissolved in ethanol (12498740, Thermo Fisher Scientific, Waltham, MA, USA), with the final ethanol concentration not exceeding 0.008% (*v*/*v*). For treatment with the AR antagonist enzalutamide (HY-70002, MedChemExpress, Monmouth Junction, NJ, USA), the compound was dissolved in dimethyl sulfoxide (DMSO; 196055, MP Biomedicals, Solon, OH, USA), ensuring that the final DMSO concentration did not exceed 0.17% (*v*/*v*).

All cell-based experiments received approval from the Ethics Committee Research UZ/KU Leuven (S66018).

### 2.2. Proteomics Analysis

Cells were lysed using radioimmunoprecipitation assay (RIPA) buffer (R0278, Sigma-Aldrich) following the manufacturer’s guidelines. Protein concentration was determined using the Pierce bicinchoninic acid protein assay kit (23227, Thermo Fisher Scientific). Fifty micrograms of protein from each sample were processed for liquid chromatography (LC)/mass spectrometry (MS) analysis using the S-Trap micro sample prep kit (K02-micro-160, ProtiFi, Fairport, NY, USA) according to the manufacturer’s instructions. Briefly, samples were solubilized in 5% (*w*/*v*) sodium dodecyl sulfate (included in the sample prep kit), reduced by 5 mM tris(2-carboxyethyl)phosphine (75259, Sigma-Aldrich), alkylated by 20 mM iodoacetamide (I1149, Sigma-Aldrich), and denatured further with 2.5% (*v*/*v*) phosphoric acid (452289, Sigma-Aldrich). Subsequently, samples were applied to the S-trap columns for purification, and proteins were digested with trypsin (V511A, Promega, Madison, WI, USA; 1 µg per 50 µg protein). Peptides were eluted sequentially using three different elution buffers: (1) 50 mM triethylammonium bicarbonate (90114, Thermo Fischer Scientific), (2) 0.2% (*v*/*v*) formic acid (84865.260, VWR, Leuven, Belgium), and (3) 50% (*v*/*v*) acetonitrile (83640.320, VWR). The eluted peptides were completely dried using speed vacuum centrifugation.

The tryptic peptides were reconstituted in solvent A (H_2_O, 0.1% (*v*/*v*) formic acid). Approximately 200 ng of the digests were loaded onto an Evotip Pure tip (Evosep, Odense, Denmark). Peptide mixtures were analyzed using an Evosep One LC (Evosep) coupled to a ZenoTOF 7600 mass spectrometer (SCIEX, Framingham, MA, USA) equipped with an OptiFlow V source (SCIEX). The method employed on the Evosep One was 30SPD (samples per day), using a 44-min gradient with the EV1137 performance column (15 cm × 150 µm, 1.5 µm). The mass spectrometer operated in positive mode with a spray voltage set to 4500 V and relied on the sequential window acquisition of all theoretical mass spectra (SWATH) mode. The SWATH acquisition scheme included 85 variable-size windows covering a precursor mass range of 350–1250 *m*/*z* with an accumulation time of 0.02 s.

All raw data from the ZenoTOF 7600 system (.wiff) were acquired using SCIEX OS software 3.3. and processed with data-independent acquisition by neural networks (DIA-NN 1.8.1) [31] in a library-free mode. Spectra were compared against the human reference proteome database (https://www.uniprot.org/proteomes/UP000005640; accessed on 10 May 2024) with the following main settings: full tryptic digest allowing up to two missed cleavage sites, variable modifications including oxidized methionines and N-terminus acetylation, and fixed modification of carbamidomethylated cysteines. In addition, the “match-between-runs” option was enabled. The neural network classifier operated in single-pass mode, and protein inference was gene-based. Quantification was performed using robust LC (high precision). Cross-run normalization was retention time-dependent, and library generation used smart profiling. The DIA-NN protein group output was further evaluated using in-house R scripts.

### 2.3. Plasmids and Electroporation

The constructs encoding peroxisomal (po-) roGFP2 [32], cytosolic (c-) roGFP2 [32], mitochondrial (mt-) roGFP2 [32], po-roGFP2-Orp1 [33], c-roGFP2-Orp1 [33], or mt-roGFP2-Orp1 [33] have been detailed elsewhere. Plasmids were introduced into the cells by electroporation using the Neon Transfection System (Thermo Fischer Scientific) along with a homemade sucrose-based buffer [34]. Electroporation parameters were as follows: RWPE-1 and PC3 (1250 V, 30-ms pulse width, 1 pulse), 22Rv1 (1250 V, 20-ms pulse width, 2 pulses), and LNCaP (1200 V, 20-ms pulse width, 2 pulses).

### 2.4. SDS-PAGE and Immunoblotting

To process the cells for SDS-PAGE and immunoblotting (IB), they were rinsed with ice-cold Dulbecco’s phosphate-buffered saline (DPBS) (BE17-512F, Lonza) and then lysed on ice for 5 min using RIPA buffer, supplemented with a protease inhibitor cocktail (P2714, Sigma-Aldrich). Subsequently, the lysates were collected and sonicated (UP50H, Hielscher, Berlin, Germany) in three 5 s cycles (settings: cycle: 0.5; amplitude: 50%) and then cleared by centrifugation at 10,000× *g* for 10 min. The supernatants were processed for reducing SDS-PAGE and IB, as previously described [33]. For samples derived from subcellular fractions, proteins were first precipitated using 6% (*w*/*v*) trichloroacetic acid (T9159, Sigma-Aldrich) and 0.0125% (*w*/*v*) sodium deoxycholate (30970, Fluka Biochemika, Buschs, Switzerland), followed by washing with acetone (20063.365, VWR). Total protein staining with Ponceau S (0.1% (*w*/*v*); 161470250, Acros Organics, Geel, Belgium) in 1% (*v*/*v*) acetic acid (20104.298, VWR) was employed as a loading control. Signal intensities of bands or lanes were quantified using ImageStudio Lite 5.2 (LI-COR Biosciences, Lincoln, NE, USA).

### 2.5. Antibodies

The following primary antibodies were used for either IB or immunofluorescence (IF): rabbit anti-3-ketoacyl-CoA thiolase 1 (ACAA1) (HPA007244, Atlas Antibodies, Stockholm, Sweden; IB, 1:1000), rabbit anti-AR (ab105225, Abcam, Cambridge, UK; IB, 1:1000; IF, 1:150), rabbit anti-CAT (219010, Calbiochem, Darmstadt, Germany; IB, 1:2000; IF, 1:200), rabbit anti-glyceraldehyde 3-phosphate dehydrogenase (GAPDH) (14C10, Cell Signaling Technology, Danvers, MA, USA; IB, 1:2000), mouse anti-glutathione reductase (GSR) (sc-133245, Santa Cruz Biotechnology, Heidelberg, Germany; IB, 1:2000), rabbit anti-hydroxysteroid 17-beta dehydrogenase 4 (HSD17B4) (15116-1-AP, Fisher Scientific, Dilbeek, Belgium; IB, 1:1000), rabbit anti-GSTP1 (354212, Calbiochem; IB, 1:2000), rabbit anti-peroxin 13 (PEX13) (IB, 1:3000) [35], rabbit anti-peroxin 14 (PEX14) (IB, 1:5000) [36], mouse anti-PEX14 (IF, 1:200) [37], rabbit anti-peroxiredoxin 2 (PRDX2) [38] (IB, 1:4000), rabbit anti-alpha tubulin (TUBA) (SC-5546, Santa Cruz Biotechnology; IB, 1:5000), rabbit anti-prostate-specific antigen (PSA) (10679-1-AP, Proteintech, Rosemont, IL, USA; IB, 1:1000), and rabbit anti-thioredoxin (TXN) (HPA047478, Sigma-Aldrich; IB, 1:2000). The following alkaline phosphatase-conjugated secondary antibodies were used for IB: anti-rabbit (A3687, Sigma-Aldrich; 1:5000) and anti-mouse (A2429, Sigma-Aldrich; 1:10,000). For IF, the secondary antibodies used were Alexa Fluor 488-conjugated anti-mouse (A11017, Invitrogen, Merelbeke, Belgium; 1:2000) and Texas Red-conjugated anti-rabbit (401355, Calbiochem; 1:200).

### 2.6. Catalase Activity

The CAT activity assay was conducted using a modified version of the method described by Baudhuin and colleagues [39]. Briefly, cells were harvested, washed with ice-cold DPBS, lysed using 2% (*w*/*v*) Triton X-100 (T9284, Sigma-Aldrich), and then mixed with either assay blank buffer (at room temperature) or assay substrate buffer (at 0 °C) for 8 min. The assay blank buffer comprised 0.25 M sucrose (220900010, Acros Organics), 20 mM imidazole (56750, Fluka; pH 7.0), 0.1% (*w*/*v*) bovine serum albumin (A7638, Sigma-Aldrich), and 0.1% (*v*/*v*) ethanol (12498740, Thermo Fisher Scientific). The assay substrate buffer was prepared by adding H_2_O_2_ (0.006% (*w*/*v*); CL00.2306.1000, Chem-lab, Zedelgem, Belgium) to the assay blank buffer. The remaining H_2_O_2_ was then quantitatively determined by adding 0.6 volumes of 14 mM titanyl sulfate (14023, Riedel-de Haën, Seelze, Germany; in 2 N sulfuric acid, 133610025, Acros Organics) and measuring the absorbance of the titanium-peroxide complex at 405 nm. One unit of CAT activity is defined as the amount of enzyme that degrades 90% of the substrate under these conditions in a volume of 50 mL. CAT activity was normalized to the protein content of each sample. To inhibit CAT, cells were treated with 10 mM 3-amino-1,2,4-triazole (3-AT) (264571000, Thermo Fisher Scientific; in water).

### 2.7. Subcellular Fractionation

RWPE-1, 22Rv1, LNCaP, and PC3 cells (50 million cells per condition) were harvested, washed with DPBS, and subsequently washed in homogenization medium composed of 250 mM sucrose, 5 mM 3-(N-morpholino)propanesulfonic acid (pH 7.2; M1254, Sigma-Aldrich), 1 mM ethylenediaminetetraacetic acid (pH 7.2; 147855000, Acros Organics), 1 mM dithiothreitol (1008, Gerbu Biotechnik, Heidelberg, Germany), 0.1% (*v*/*v*) ethanol (E/0650DF/15, Fisher Chemical, Pittsburgh, PA, USA), and a protease inhibitor cocktail (P2714, Sigma-Aldrich). The cell pellet was homogenized in this medium using a stainless-steel tissue grinder (885310-0007, Kontes, Millville, NJ, USA). Approximately 20 strokes were sufficient to disrupt the cell membrane without affecting the subcellular organelles. All reagents were pre-cooled to minimize proteolysis. The total homogenate (H) was subjected to differential centrifugation to separate subcellular fractions. Initially, the homogenate was centrifuged at 1300× *g* for 20 min to obtain the nuclear (N) and post-nuclear (PNS) fractions. Subsequently, the PNS was centrifuged at 2330× *g* for 10 min to pellet the heavy mitochondrial fraction (M). The resulting supernatant was then centrifuged at 13,000× *g* for 20 min to pellet the light mitochondrial fraction (L), which is enriched in peroxisomes. Finally, the remaining supernatant was centrifuged at 100,000× *g* for 60 min to yield the cytosolic (S) and microsomal (P) fractions.

### 2.8. Fluorescence Microscopy and Immunofluorescence

Cells were fixed for IF microscopy and processed as described elsewhere [40]. Fluorescence microscopy analysis was conducted as previously outlined [32,41], using the following filter cubes: F400 (excitation: 390–410 nm; dichroic mirror: 505 nm; emission: 510–550 nm), F480 (excitation: 470–495 nm; dichroic mirror: 505 nm; emission: 510–550 nm), and a Texas Red filter cube (excitation: 562.5/35 nm; dichroic mirror: 600 nm; emission: 610 nm long pass). Cells were seeded and imaged in FluoroDish cell culture dishes (FD-35, World Precision Instruments, Hertfordshire, UK) pre-coated with 25  μg/mL polyethyleneimine (195444, MP Biomedicals) [33]. Image acquisition and analysis were performed using cellSens Dimension software (version 2.1) (Olympus, Puurs-Sint-Amands, Belgium).

### 2.9. Proliferation and Sulforhodamine B Assays

Proliferation assays were conducted using the IncuCyte S3 imaging system (Sartorius, Göttingen, Germany). Cells were seeded in IncuCyte Imagelock 96-well plates (Sartorius), which were pre-coated with 25  μg/mL polyethyleneimine and placed in an IncuCyte^®^ S3 live-cell imaging system set to 37 °C in a humidified atmosphere containing 5% CO_2_. Phase contrast images of the wells were captured continuously for at least 70 h. The IncuCyte S3 2022A Rev1 software was used to calculate the percentage of cell confluence in each well at all time points.

For the sulforhodamine B assay, cells were seeded into pre-coated 96-well plates. Six hours later, designated as the starting point, the medium was removed from the wells, and the cells were washed with DBPS and fixed with 4% (*w*/*v*) paraformaldehyde (P6148, Sigma-Aldrich; pH: 7.2–7.4) for 20 min. The cells were then incubated with sulforhodamine B dye (341738, Sigma-Aldrich; 0.057% (*w*/*v*) in 1% (*v*/*v*) acetic acid) for 30 min. Excess dye was washed off using 1% (*v*/*v*) acetic acid, and the fixed cells were incubated for 5 min with 10 mM Tris(hydroxymethyl)aminomethane (Tris base; T1378, Sigma-Aldrich; pH: 10.5) to release the dye for optical density measurement at 510 nm. After three days, the same steps were repeated for the remaining wells, and the optical density data were averaged and compared to the starting time point.

### 2.10. Knockdown of Catalase Expression

To transiently knock down CAT expression, two independent Dicer-substrate small interfering (Dsi-)RNAs (Integrated DNA Technologies, Leuven, Belgium) were used: (i) hs.Ri.CAT.13.10 (sense strand: 5′-rArUrCrArArArArArCrCrUrUrUrCrUrGrUrUrGrArArGrATG-3′; antisense strand: 5′-rCrArUrCrUrUrCrArArCrArGrArArArGrGrUrUrUrUrUrGrArUrGrC-3′) and (ii) hs.Ri.CAT.13.20 (sense strand: 5′-rCrCrGrUrCrArUrGrGrCrUrUrArArUrGrUrUrUrArUrUrCCT-3′; antisense strand: 5′-rArGrGrArArUrArArArCrArUrUrArArGrCrCrArUrGrArCrGrGrUrG-3′). LNCaP and 22Rv1 cells were reverse transfected with a final concentration of 50 to 200 nM using Lipofectamine 3000 (L3000015, Invitrogen). Four days later, the cells were forward transfected and then either (i) processed for proliferation analysis or (ii) seeded into 6-well plates and processed for IB on the third day. A negative control DsiRNA (51-01-14-03, Integrated DNA Technologies), which does not target any sequences in the human transcriptomes, was included. Transfection efficiency was evaluated using a fluorescent TYE 563-labeled Transfection Control DsiRNA (51-01-20-19, Integrated DNA Technologies) and found to be nearly 100%.

### 2.11. Statistical Analysis

Data are expressed as mean ± standard deviation. Statistical analysis was conducted based on the distribution and nature of the collected data. For comparisons among multiple groups, a one-way or two-way analysis of variance (ANOVA) (followed by Tukey’s or Dunnett’s tests), a nested ANOVA, or the Kruskal–Wallis test (followed by a Dunn’s test) was used. For comparisons between two groups, an unpaired two-tailed *T*-test, a nested *T*-test, or a Mann–Whitney test was used. GraphPad Prism Software (version 9.4) (San Diego, CA, USA) was employed for analysis. A *p*-value less than 0.05 was considered statistically significant. For the proteomics data, FragPipe-Analyst (http://fragpipe-analyst.nesvilab.org/; accessed on 13 August 2024) was used to perform statistical and principal component analysis (data type: DIA; *p*-value cut-off: 0.05; log_2_ FC cut-off: 1). *p*-values were adjusted using the Benjamini–Hochberg method to correct for the false discovery rate (FDR). In the R1881 proteomics analysis, missing values were imputed using the *k*-nearest neighbors (KNN) method. The Kyoto Encyclopedia of Genes and Genomes (KEGG) enrichment analysis was performed using ShinyGO 0.80 (http://bioinformatics.sdstate.edu/go/; accessed on 13 August 2024), with the following settings: selected species, human; FDR cut-off, 0.05; sort by FDR; the total list of identified proteins was uploaded as “background”.

## 3. Results

### 3.1. Comparative Analysis of the Peroxisomal, Mitochondrial, and Cytosolic Redox Profiles in Prostate Cell Lines

To obtain an initial understanding of the redox profile in peroxisomes across different prostate cell lines, we analyzed the ratio of oxidized glutathione (GSSG) to reduced glutathione (GSH) (Figure 1A) and measured the relative levels of H_2_O_2_ (Figure 1B) within peroxisomes. These results were then compared with similar measurements from mitochondria and the cytosol. Our findings revealed that culturing RWPE-1 cells in either KSFM or MEM only affected the peroxisomal H_2_O_2_ levels. In addition, when cultured in MEM, (i) PCa cells showed a more reduced glutathione redox state in the cytosol compared to RWPE-1 cells, while cytosolic H_2_O_2_ levels remained unchanged; (ii) LNCaP cells exhibited lower peroxisomal H_2_O_2_ levels compared to RWPE-1 cells; and (iii) although mitochondrial glutathione levels did not differ significantly between malignant and non-malignant cells, H_2_O_2_ levels were lower in LNCaP and PC3 cells compared to RWPE-1 cells. Representative images of the subcellular distribution patterns of the fluorescent reporter proteins are shown in Appendix A.

### 3.2. Unbiased Proteomics Reveals Differential Peroxisomal and Redox Proteomes in Benign and Malignant Prostate Cell Lines

To investigate differences in the peroxisomal and redox proteomes between the benign prostate cell line (RWPE-1; cultured in KSFM or MEM) and the malignant prostate cell lines (22Rv1, LNCaP, and PC3; cultured in MEM), we conducted an unbiased proteomics analysis on total cell lysates. We identified 7502 proteins (6807 in RWPE-1/KSFM, 6923 in RWPE-1/MEM, 6762 in 22Rv1/MEM, 6613 in LNCaP/MEM, and 6923 in PC3/MEM) across all conditions (Appendix A). To validate this data set, we performed an IB analysis for four peroxisome-related proteins (ACAA1, AMACR, PEX13, and PEX14) (Appendix A) and four antioxidant enzymes (GSR, GSTP1, PRDX2, and TXN) (Appendix A). The protein abundances measured by both proteomics and IB analyses were consistent for all tested proteins.

To simplify and identify trends in this extensive data set, we carried out a principal component analysis on all samples (Figure 2). This procedure revealed that (i) malignant and non-malignant cells clustered differently, (ii) there was a clear proteomic difference between AR-positive (22Rv1 and LNCaP) and AR-negative (PC3) PCa cells, (iii) AR-positive PCa cells exhibited similar clustering patterns, and (iv) switching the medium from KSFM to MEM had only a minor impact on the proteome of RWPE-1 cells.

Figure 3A displays the number of proteins that are significantly upregulated, downregulated, or unchanged in each comparison. To identify enriched pathways, we performed KEGG enrichment analysis on proteins with at least a 2-fold significant change in expression (Figure 3B for upregulated proteins; Figure 3C for downregulated proteins). Our analysis revealed that the “peroxisome” pathway was notably prominent among the highly expressed proteins in the AR-positive PCa cell lines (22Rv1 and LNCaP) (Figure 3B).

The proteomics experiments identified 76 peroxisome-related proteins across all conditions. Combined with the KEGG analysis, this led us to focus on differences in peroxisomal proteins. Consequently, we created a heatmap that highlights the differentially expressed peroxisomal proteins in PCa cells compared to RWPE-1 cultured in the same medium (Figure 4). Given the versatile nature of peroxisomes, we classified the identified peroxisomal proteins based on their function. We observed significant differences in peroxisomal protein expression between non-malignant and malignant prostate cells. Specifically, CAT, bifunctional epoxide hydrolase 2 (EPHX2), and glutathione S-transferase kappa 1 (GSTK1) exhibited higher expression across all PCa cell lines, whereas peroxiredoxin 5 (PRDX5) levels were reduced. Similarly, the peroxisomal fatty acid oxidation enzymes ACAA1, enoyl-CoA delta isomerase 2 (ECI2), AMACR, and trans-2-enoyl-CoA reductase (PECR) were consistently upregulated. Furthermore, increased expression was observed for peroxin 26 (PEX26), a protein involved in peroxisomal protein import, as well as ganglioside-induced differentiation-associated protein 1 (GDAP1), a peroxisomal fission factor.

To gain a clearer view of the antioxidant enzyme profiles in the different cell lines, we also visualized these proteins using a heatmap (Figure 5). Along with the previously discussed changes in peroxisomal antioxidant enzymes, we observed a notable increase in the levels of GSR, maleylacetoacetate isomerase (GSTZ1), and PRDX2, and a decrease in glutathione peroxidase 4 (GPX4), mitochondrial glutathione S-transferase 3 (MGST3), glutathione S-transferase omega1 (GSTO1), TXN, glutathione peroxidase 1 (GPX1), microsomal glutathione S-transferase 1 (MGST1), and GSTP1 in PCa cells compared to the normal prostate cell line. Note that the GSTP1 promoter is frequently hypermethylated in PCa tissue specimens [8]. Although these proteomics data provide valuable insights (see Discussion), they do not fully capture the complex changes in redox state at the subcellular level (Figure 1). This limitation is likely due to the multifaceted nature of redox homeostasis across different subcellular compartments and the critical role of post-translational modifications in regulating antioxidant enzyme activity [42,43].

### 3.3. Expression, Activity, and Localization of Catalase in Prostate Cancer Cell Lines

Considering this study’s focus on the role of peroxisomes in PCa pathogenesis, it is noteworthy that CAT, a key peroxisomal antioxidant enzyme known for efficiently scavenging excess H_2_O_2_, consistently showed higher expression levels in malignant prostate cells compared to non-malignant cells (Figure 6A,B). However, its enzymatic activity was significantly lower in PC3 cells compared to 22Rv1 and LNCaP cells (Figure 6C).

Because a protein’s location can significantly affect its role in cellular processes, and since CAT is present in both peroxisomes and the cytosol, confirming its presence in peroxisomes was crucial. IF imaging showed that CAT colocalized with the peroxisomal marker PEX14 in all cell lines (Appendix A). However, distinguishing the CAT signal in the cytosol was challenging due to background fluorescence and variations in cell thickness. To address this issue, we conducted subcellular fractionation to separate peroxisomes (L fraction) from the cytosol (S fraction). We used HSD17B4, a peroxisomal matrix protein, and PEX14, a peroxisomal membrane protein, as markers to verify accurate fractionation and minimize any effects of peroxisomal matrix protein leakage during the procedure. TUBA served as a cytosolic marker (Appendix A). Despite the upregulation of CAT in PCa cells and its relatively inefficient targeting to peroxisomes [18], our results indicated that the distribution of CAT between peroxisomes and the cytosol was uniform across all tested cell lines and culture conditions (Appendix A).

### 3.4. Chemical Inhibition of Catalase Activity Reduces LNCaP Cell Proliferation

To determine whether CAT activity contributes to the proliferation of PCa cells, we initially used the CAT inhibitor 3-AT. We first verified that 10 mM 3-AT effectively inhibited CAT activity in all cell lines (Figure 7A). Notably, 3-AT treatment significantly suppressed the proliferation of LNCaP cells but had little to no inhibitory effect on the proliferation of PC3 and 22Rv1 cells (Figure 7B). Interestingly, 3-AT also significantly inhibited the proliferation of the control prostate cell line RWPE-1 cultured in KSFM but not in MEM (Figure 7B). However, given that RWPE-1 cells exhibit reduced growth in MEM, it is not surprising that no growth inhibition was observed under these conditions.

### 3.5. Knockdown of Catalase Expression Stimulates the Proliferation of LNCaP and 22Rv1 Cells

To independently verify that the reduction in LNCaP cell proliferation was due to CAT activity inhibition, we performed a transient knockdown of CAT using two different DsiRNAs targeting distinct regions of the CAT transcript (Figure 8B). Contrary to our results with 3-AT, we observed an increase in LNCaP cell proliferation instead of a decrease (Figure 8A). These findings were confirmed by a sulforhodamine B cell viability assay (Appendix A). In addition, the extent of proliferation depended on the concentration of the CAT DsiRNA used (Appendix A) and was also observed in 22Rv1 cells (Figure 8A). A plausible explanation for the seemingly contradictory 3-AT and CAT DsiRNA data is provided in the discussion.

### 3.6. Androgen Receptor Activation by R1881 Alters Peroxisomal Redox State

To investigate how AR activation affects subcellular redox status, we analyzed the glutathione redox couple and H_2_O_2_ levels in LNCaP cells treated with the synthetic androgen R1881. R1881 is known to have biphasic effects on LNCaP cellular proliferation: it stimulates proliferation at lower concentrations but inhibits it at higher concentrations [44,45,46]. Our proliferation assays confirmed these effects, showing stimulation at 1 pM R1881 and inhibition at 0.1, 1, and 10 nM concentrations (Appendix A). Consequently, we assessed the redox status at 1 pM and 10 nM R1881. At 1 pM, R1881 did not alter the redox state in any of the tested compartments (Figure 9A). In contrast, at 10 nM, we observed increased H_2_O_2_ levels and a reduced glutathione redox state in peroxisomes, along with elevated levels of oxidized glutathione in mitochondria, compared to controls (Figure 9B).

### 3.7. R1881 Modulates the Peroxisomal and Antioxidant Enzyme Proteome

To explore the potential mechanisms behind the increased peroxisomal H_2_O_2_ levels observed after R1881 treatment, we examined how activation of AR affected the abundance of antioxidant enzymes and peroxisome-associated proteins. Proteomics analysis of LNCaP cells treated or not with 10 nM R1881 identified 6810 proteins (Appendix A). Among these, 391 were significantly upregulated (FC ≥ 2), and 429 proteins were significantly downregulated (FC ≥ 2) (Figure 10A). Molecular Signatures Database (MsigDB) hallmarks and KEGG enrichment analyses of the differentially expressed proteins revealed that the “Androgen response” hallmark was the most enriched (Figure 10B). In addition, genes regulated by the E2F family of transcription factors, which are critical for processes like the cell cycle and DNA replication, were enriched among the downregulated proteins (Figure 10C). The volcano plot highlights some examples of upregulated androgen-responsive proteins and downregulated proteins involved in the cell cycle, DNA replication, and cell division (Figure 10A).

Treatment with R1881 resulted in increased levels of acyl-coenzyme A oxidase 1 (ACOX1) and ACOX3, two enzymes that produce H_2_O_2_ in peroxisomes. Concurrently, R1881 treatment decreased the levels of CAT, peroxiredoxin 1 (PRDX1), and mitochondrial amidoxime reducing component 2 (MARC2) (Figure 11), which are involved in H_2_O_2_ breakdown and are found in both peroxisomes and other subcellular compartments [14]. Consistent with these findings, R1881 also reduced CAT activity (Figure 12). These changes likely contribute to the elevated H_2_O_2_ levels observed in peroxisomes following treatment. Furthermore, R1881 affected the expression levels of several enzymes involved in either the consumption of GSH (e.g., GPX1, GPX4, and GPX8) or the regeneration of GSSG (e.g., GSR) (Figure 13). However, the trends observed in the proteomics data do not provide sufficient information to explain why the glutathione redox state is more reduced in peroxisomes and more oxidized in mitochondria.

Treating LNCaP cells with the AR antagonist enzalutamide did not significantly affect catalase activity or subcellular H_2_O_2_ levels (Appendix A). This lack of phenotypic change is consistent with transcriptome data indicating that enzalutamide treatment results in only a slight, non-significant increase in CAT levels (Appendix A) [47], likely due to the already low androgen levels in the culture medium [48]. In addition, reducing catalase levels did not impact AR expression or activity, as demonstrated by prostate-specific antigen (PSA) staining (Appendix A). These observations suggest that CAT likely functions downstream of the AR signaling pathway rather than upstream.

## 4. Discussion

In this study, we combined subcellular measurements of the glutathione redox state and H_2_O_2_ levels with a comprehensive proteomics analysis to compare the peroxisomal and redox profiles of one benign and three malignant prostate cell lines. Our data indicate that the peroxisome pathway, including CAT expression and activity, is significantly upregulated in AR-positive cell lines. The significance of these findings, along with other relevant elements, is critically discussed and contextualized in the following sections.

Initially, we compared the subcellular glutathione redox state and H_2_O_2_ levels between non-malignant and malignant prostate cell lines. Our observations revealed that (i) the cytosolic glutathione redox state was more reduced in PCa cells compared to the benign prostate cell line (Figure 1A) and (ii) H_2_O_2_ levels were lower in the peroxisomes and mitochondria of LNCaP cells, with mitochondrial H_2_O_2_ levels also reduced in PC3 cells (Figure 1B). These findings align with our proteomics data, which show significant changes in proteins involved in glutathione metabolism and H_2_O_2_ degradation in PCa cells. For example, the combined abundance of GSH-consuming enzymes listed in Figure 5 decreased 7.7-, 3.3-, and 3.1-fold in 22Rv1, LNCaP, and PC3 cells, respectively (Appendix A). Conversely, the expression levels of GSR, a primarily cytosolic enzyme that regenerates GSH from GSSG using NADPH, along with the combined expression levels of NADPH-generating enzymes (IDH1, IDH2, PGD, and G6PD), were upregulated 1.3 to 2.4-fold in these PCa cell lines (Appendix A). These findings are consistent with a previous study indicating that human prostate carcinoma cell lines exhibit significantly lower intracellular GSSG/GSH ratios compared to normal prostate epithelial cells [49]. Regarding H_2_O_2_ degradation, our data highlight several key points: (i) the protein expression levels and activity of CAT are upregulated in the AR-positive 22Rv1 and LNCaP PCa cell lines (Figure 6), (ii) SOD2, an H_2_O_2_-producing enzyme primarily located in the mitochondrial matrix, is downregulated in LNCaP and PC3 cells, and (iii) MARC2, another H_2_O_2_-degrading enzyme found in mitochondria and peroxisomes, is upregulated in LNCaP and PC3 cells (Figure 5).

At first glance, our observation that CAT levels are elevated in PCa cell lines appears to contradict findings from patient-derived prostate adenocarcinoma samples, which report reduced CAT levels [20,21,22] and increased H_2_O_2_ levels [9,10]. However, the small sample sizes and the focus on localized disease in these studies may limit their comparability to our cell lines, which are derived from late-stage disease. In addition, these discrepancies may arise from differences between in vitro culture conditions (e.g., oxygen tension, nutrient composition) and the actual tumor microenvironment. Our data showing that AR activation reduces CAT levels and activity in LNCaP cells (Figure 11 and Figure 12) suggest that the differences in CAT levels between cultured cells and patient prostate tissue may be attributed to the significantly higher androgen levels present in prostate cancer tissue compared to standard cell culture media [49]. Supporting this notion, another study found no differences in CAT expression between primary cultures of prostatic epithelial cells from malignant and non-diseased tissues [50]. Furthermore, it is important to recognize that catalase expression can be regulated by various transcription factors, as well as through genetic, epigenetic, and post-transcriptional processes [51]. One example is the AR-mediated suppression of CAT via FOXO3A-dependent signaling [52].

It is crucial to consider that (i) data from clinically heterogeneous bulk tissue may not accurately reflect the expression levels observed in cell type-resolved analyses [53,54], (ii) there is a general lack of comparative proteomics data between normal and primary prostate adenocarcinoma tissue in publicly accessible cancer omics databases, such as the University of Alabama at Birmingham Cancer Data Analysis Portal (UALCAN; https://ualcan.path.uab.edu/; accessed on 13 August 2024) [55], and (iii) this study is the first to provide detailed information about the proteome of commonly used prostate cell lines. Although mRNA data are available for patients with different Gleason scores, as illustrated for catalase (Appendix A), these values, while informative and valuable, do not always accurately reflect protein abundance. Factors such as post-transcriptional regulation, translation efficiency, protein stability, and feedback mechanisms can lead to discrepancies, rendering transcriptomics analyses insufficient on their own. However, data from the Human Protein Atlas database (HPA; https://www.proteinatlas.org/; accessed on 19 August 2024) provide comparative antibody staining information across various cancer tissues [56], generally supporting our findings in PCa cell lines: GSTP1 staining is almost always negative in prostate cancer samples; CAT and MARC2 show moderate staining in 10 to 40% of samples; and GSR, G6PD, and IDH1 exhibit moderate to strong positivity in most of the samples tested.

To further explore how variability in CAT expression among different prostate cancers affects PCa cell proliferation, we initially used 3-AT (10 mM) to inhibit CAT activity. This treatment significantly reduced the proliferation of LNCaP cells but had minimal effects in 22Rv1 and PC3 cells (Figure 7B). While 3-AT effectively inhibited CAT activity in all cell lines (Figure 7A), another study reported that a much higher concentration of 3-AT (250 mM) inhibited PC3 cell proliferation [57]. Notably, the knockdown of CAT using DsiRNAs resulted in increased proliferation of both LNCaP and 22Rv1 cells (Figure 8A,B). Given that reduced CAT activity can lead to elevated physiological levels of H_2_O_2_, potentially activating signaling pathways that drive cancer cell metabolism and proliferation [19], these findings are not entirely surprising. In addition, they raise questions about the specificity of 3-AT as a CAT inhibitor and suggest alternative mechanisms of action. Previous studies have documented 3-AT’s ability to inhibit α-oxidation of fatty acids [58], heme synthesis [59], and cytochrome P450 2E1 activity [60].

Consistent with our findings, silencing CAT expression in human breast cancer cells led to increased H_2_O_2_ production and enhanced cell proliferation [61]. Similarly, mouse aortic endothelial cells with overexpressed CAT showed reduced proliferation rates, lower activities of cyclin D–cyclin-dependent kinase 4 and cyclin E-cyclin-dependent kinase 2 complexes, and a delayed transition from the G0/G1 phase to the S phase in the cell cycle [62]. In contrast, PC3 cells with half the normal levels of CAT exhibited decreased cellular proliferation, migration, and invasion [57]. Interestingly, although PC3 cells had higher CAT protein levels compared to RWPE-1 cells, their CAT activity was not significantly different from RWPE-1 cells (Figure 6). Therefore, we focused on CAT knockdown only in LNCaP and 22Rv1 cells, which had the highest CAT protein levels and enzymatic activity among all the cell lines (Figure 6). Our finding that reducing CAT expression stimulates LNCaP cell proliferation aligns with previous observations indicating that these cells require a prooxidant state for proliferation [49,52].

Since AR activation is a key driver of PCa and modulation of AR activity can alter the redox environment in PCa cells [63,64,65], it was important to investigate how AR activation affects the peroxisomal redox state. At a low concentration (1 pM), which stimulates cell proliferation (Appendix A), R1881 did not affect the glutathione redox state or H_2_O_2_ levels in the peroxisomal, cytosolic, or mitochondrial compartments (Figure 9A). However, at a higher concentration (10 nM), R1881 significantly increased H_2_O_2_ levels and lowered the GSSG/GSH ratio in peroxisomes (Figure 9B). These results are consistent with our findings that 10 nM R1881 downregulates CAT expression (Figure 11, Appendix A) and activity (Figure 12), while enhancing the fatty acid metabolism pathway (Figure 10), including the expression of the H_2_O_2_-producing peroxisomal β-oxidation enzymes ACOX1 and ACOX3 (Figure 11). In addition, these findings overall align with mRNA expression data from 22Rv1, VCaP, and LNCaP cells available in the Gene Expression Omnibus (GEO) database, with accession numbers GSE130534 [47], GSE220618 [66], GSE247592 [67], and GSE214756 [68] (Appendix A). Moreover, when combined with the observation that transient exposure to androgens, including synthetic R1881, can induce a quiescent state in dispersed PCa cells through redox imbalances [69], our findings offer initial molecular insights into how supraphysiological testosterone levels may influence redox imbalances in peroxisomal and mitochondrial compartments in patients undergoing bipolar androgen therapy [70]. Since elevated H_2_O_2_ levels can trigger adaptive responses, such as increased GSH levels, understanding these adaptations is crucial for explaining how PCa cells maintain redox balance, reduce their sensitivity to toxic ROS levels, and enhance their resistance to radiotherapy.

Lastly, we acknowledge several limitations of our study. First, our proteomics analyses did not identify some well-known peroxisomal proteins (e.g., D-amino acid oxidase, 2-hydroxyacid oxidase, polyamine oxidase, pipecolic acid oxidase, and sarcosine oxidase). In addition, while we observed changes in the cytosolic glutathione redox state (Figure 1) and the protein levels of enzymes involved in NADPH-dependent glutathione (GSH) regeneration and the maintenance of the NADP^+^/NADPH balance (Figure 5), we did not directly measure the levels of these pyridine nucleotides. Furthermore, since our findings are primarily based on commonly used prostate cell lines, it is important to validate these results in patient tissue samples.

## 5. Conclusions

In summary, this study demonstrates that each prostate cancer cell line exhibits unique redox characteristics and underscores the pivotal role of CAT in PCa pathogenesis by linking peroxisomal redox imbalances to cell proliferation. These findings underscore the critical role of peroxisomes in PCa biology and position CAT as a promising therapeutic target. Targeting CAT could disrupt H_2_O_2_-mediated tumor-promoting signaling pathways that drive cancer cell metabolism and proliferation [19] and increase the susceptibility of PCa cells to oxidative stress-induced cell death [57,71].

## Figures and Tables

**Figure 1 antioxidants-13-01340-f001:**
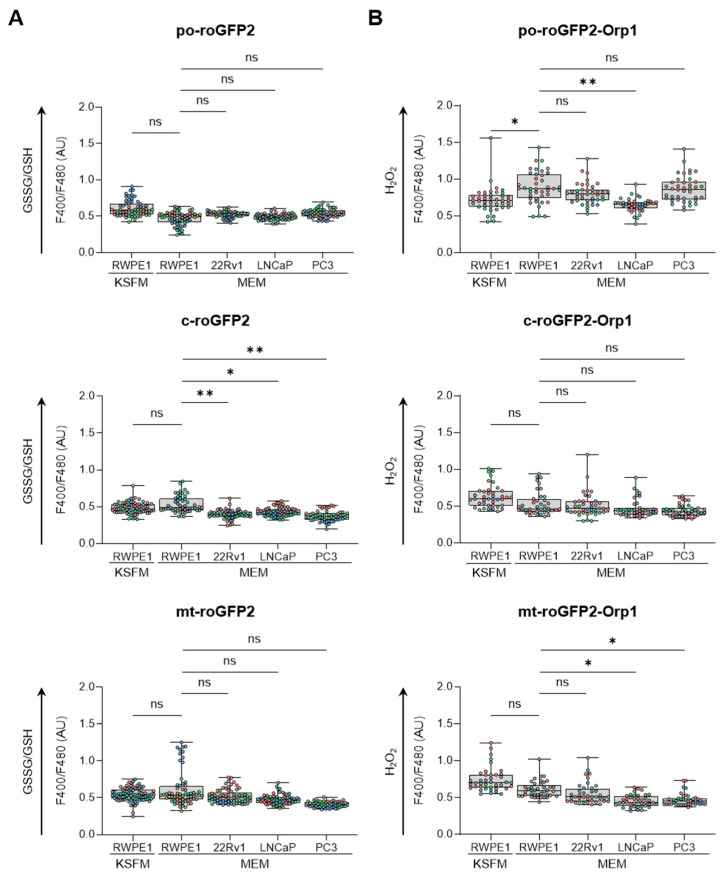
Comparison of the subcellular glutathione redox state and H_2_O_2_ levels between non-malignant and malignant prostate cell lines. RWPE1, 22Rv1, LNCaP, and PC3 cells were transfected with a plasmid encoding a peroxisomal (po-), cytosolic (c-), or mitochondrial (mt-) variant of (**A**) the glutathione redox sensor roGFP2, or (**B**) the H_2_O_2_ sensor roGFP2-Orp1. Cells were cultured in regular minimum essential medium (MEM), or for RWPE-1, also in keratinocyte serum-free medium (KSFM). One and two days later, the F400/F480 response ratios of roGFP2 and roGFP2-Orp1 were measured and presented as box plots. Each box plot shows the interquartile range, with the bottom and top edges representing the 25th and 75th percentiles, respectively. The line inside the box indicates the median, and the lines extending from the box represent one standard deviation below and above the mean. Data are derived from three independent experiments, each represented by a different color. Statistical comparisons were carried out against the RWPE-1/MEM condition using a nested one-way ANOVA (ns, *p* ≥ 0.05; *, *p* < 0.05; **, *p* < 0.01). AU, arbitrary units.

**Figure 2 antioxidants-13-01340-f002:**
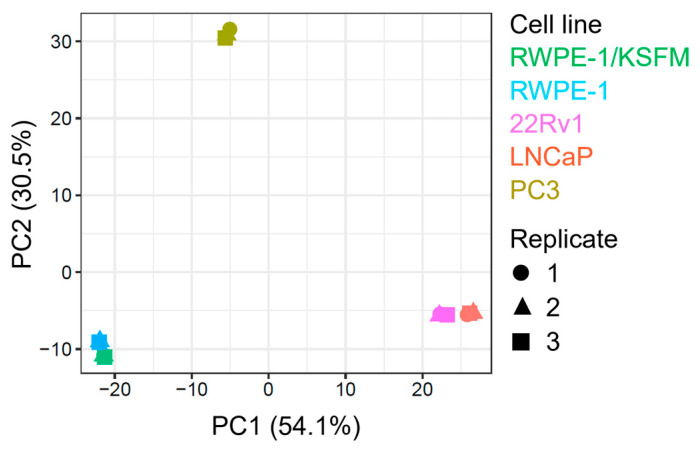
Principal component analysis of proteomics data from four prostate cell lines. The plot was generated by FragPipe-Analyst using protein abundance data from one benign (RWPE-1) and three malignant prostate cell lines, including two androgen-responsive (LNCaP and 22Rv1) and one androgen-resistant (PC3) cell line. The cells were cultured in regular MEM (not indicated) or, in the case of RWPE-1, also in KSFM. The results of three biological replicates are plotted. The x-and y-axes provide the percent variance explained by each principal component (PC).

**Figure 3 antioxidants-13-01340-f003:**
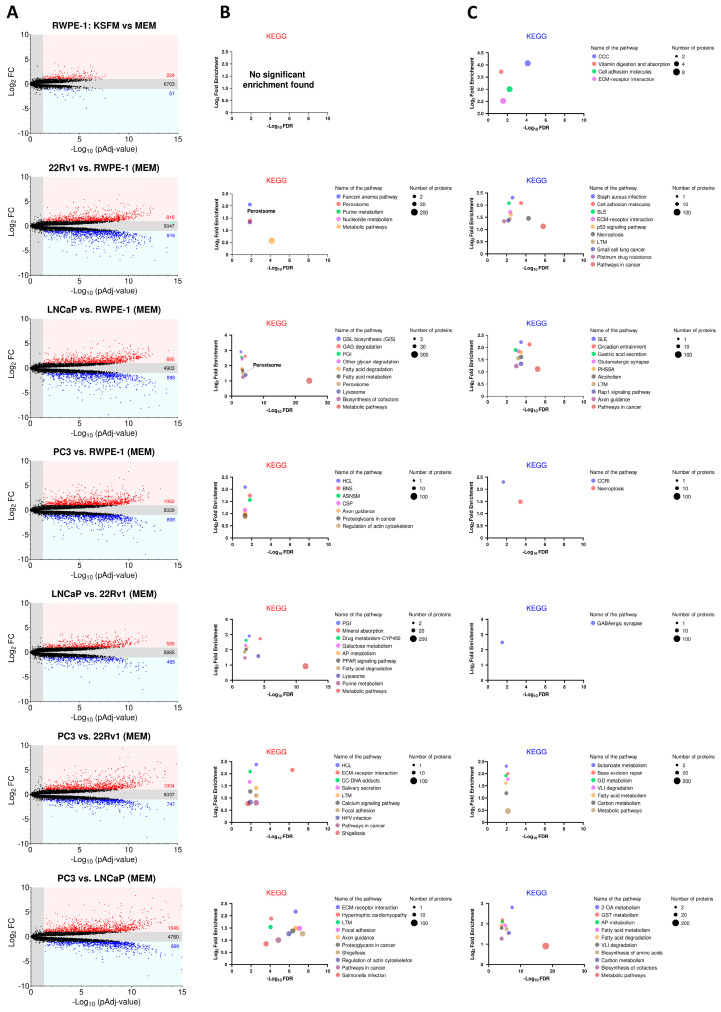
KEGG enrichment analysis of differentially expressed proteins across four prostate cell lines. (**A**) Volcano plots depicting the differences in protein abundance among cell lines. Proteins significantly upregulated (FC ≥ 2) or downregulated (FC ≤ 0.5) (pAdj < 0.05) are represented by a red or blue dot, respectively (n = 3 biological replicates). Black dots represent proteins that do not show statistically significant differences in expression. The numbers in the panels indicate the total number of proteins within each group. The list of differentially expressed proteins was generated by FragPipe-Analyst. The Benjamini–Hochberg method was used to correct for false discovery rates. (**B**,**C**) KEGG analyses of the differentially expressed proteins using ShinyGO 0.80 with the following settings: selected species, human; false discovery rate (FDR) cut-off, 0.05; sort by FDR; the total list of identified proteins was uploaded as “background”. The top KEGG pathways with significant enrichment (up to 10) are shown for proteins with (**B**) increased and (**C**) decreased expression. Pathways are sorted by fold enrichment. The size of the circles indicates the number of proteins identified in each KEGG pathway, with larger circles representing higher numbers. In panel (**B**), the circles for “Peroxisome” and “Purine metabolism” overlap in the 22Rv1 vs. RWPE-1 plot. ASNSM, amino sugar and nucleotide sugar metabolism; 2OA, 2-oxocarboxylic acid; AP, arginine/proline; BNS, biosynthesis of nucleotide sugars; CCC, complement and coagulation cascades; CC, chemical carcinogenesis; CCRI, cytokine-cytokine receptor interaction; CSP, chemokine signaling pathway; ECM, extracellular matrix; GSL, glycosphingolipid; GAG, glycosaminoglycan; GD, glyoxylate and dicarboxylate; GST, glycine/serine/threonine; HCL, hematopoietic cell lineage; HPV, human papillomavirus; LTM, leukocyte transendothelial migration; PGI, pentose and glucuronate interconversions; PHSSA, parathyroid hormone synthesis secretion and action; PPAR, peroxisome proliferator-activated receptor; SLE, systemic lupus erythematosus; VLI, valine/leucine/isoleucine.

**Figure 4 antioxidants-13-01340-f004:**
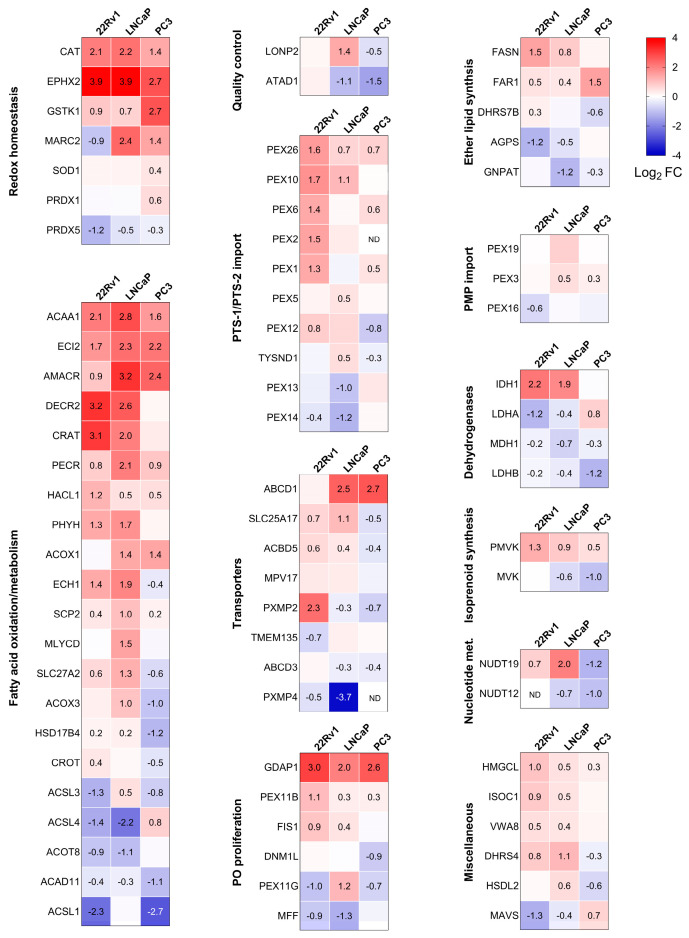
Heatmaps showing the proteome profile of 76 peroxisome-associated proteins across three distinct malignant prostate cell lines relative to RWPE-1. The relative protein abundances, expressed as Log_2_ fold change (FC), are depicted by a false color scale where red indicates upregulation, white symbolizes equal expression, and blue denotes downregulation compared to RWPE-1 cells cultured in regular MEM. Each box represents the average of three biological replicates, with Log_2_ FC displayed only if the adjusted *p*-value is <0.05. ND, protein not detected. Met, metabolism; PMP, peroxisomal membrane protein; PTS, peroxisomal targeting sequence; PO, peroxisome.

**Figure 5 antioxidants-13-01340-f005:**
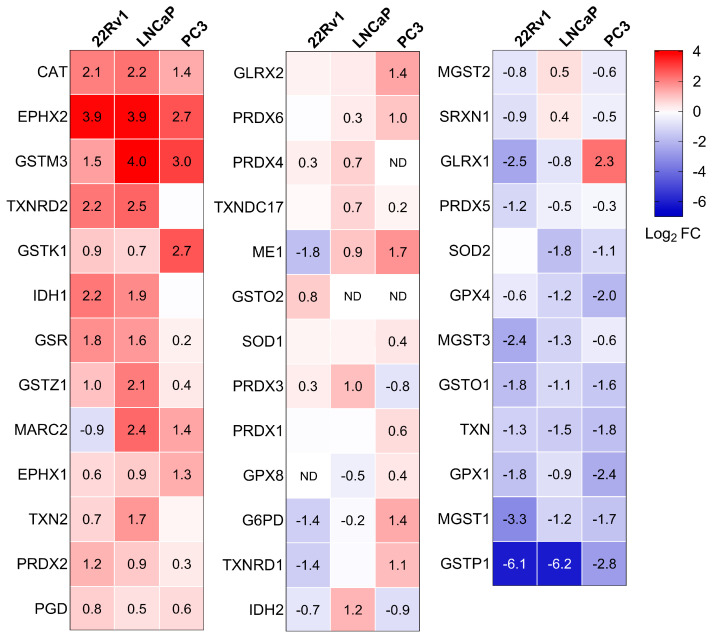
Heatmaps depicting the proteome profile of 38 primary and secondary antioxidant enzymes in three malignant prostate cell lines relative to RWPE-1. The relative protein abundances, expressed as Log_2_ FC, are depicted by a false color scale where red indicates upregulation, white symbolizes equal expression, and blue denotes downregulation compared to RWPE-1 cells cultured in regular MEM. Each box represents the average of three biological replicates, with Log_2_ FC displayed only if the adjusted *p*-value is < 0.05. ND, protein not detected.

**Figure 6 antioxidants-13-01340-f006:**
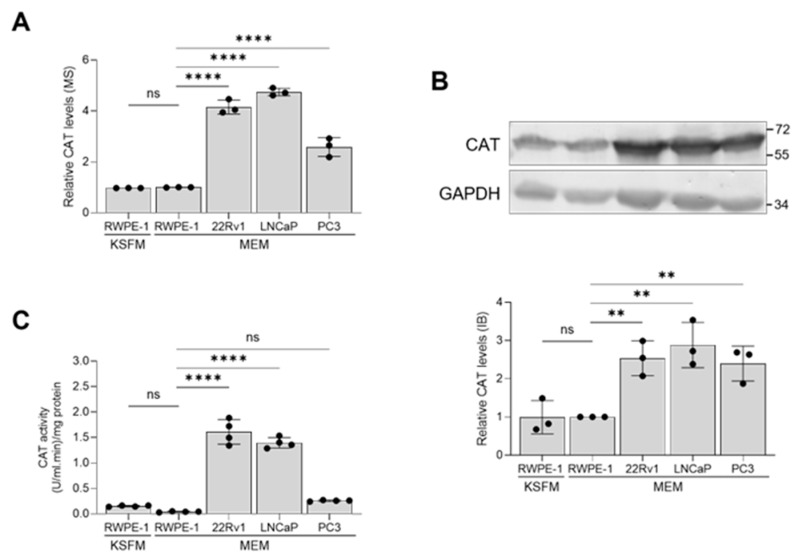
Validation of the catalase mass spectrometry data through immunoblotting and activity measurements. The relative abundance of catalase (CAT) among different cell lines was determined by (**A**) mass spectrometry and (**B**) immunoblotting. A representative blot is shown, with relevant molecular mass markers displayed on the right. Glyceraldehyde-3-phosphate dehydrogenase (GAPDH) was used as a loading control. The values are normalized to those in the RWPE-1/MEM condition, which was assigned a baseline value of 1. (**C**) CAT activity in lysates from the different cell lines. All values represent the mean ± standard deviation of 3 or 4 biological replicates. **, *p* < 0.01; ****, *p* < 0.0001; ns, non-significant.

**Figure 7 antioxidants-13-01340-f007:**
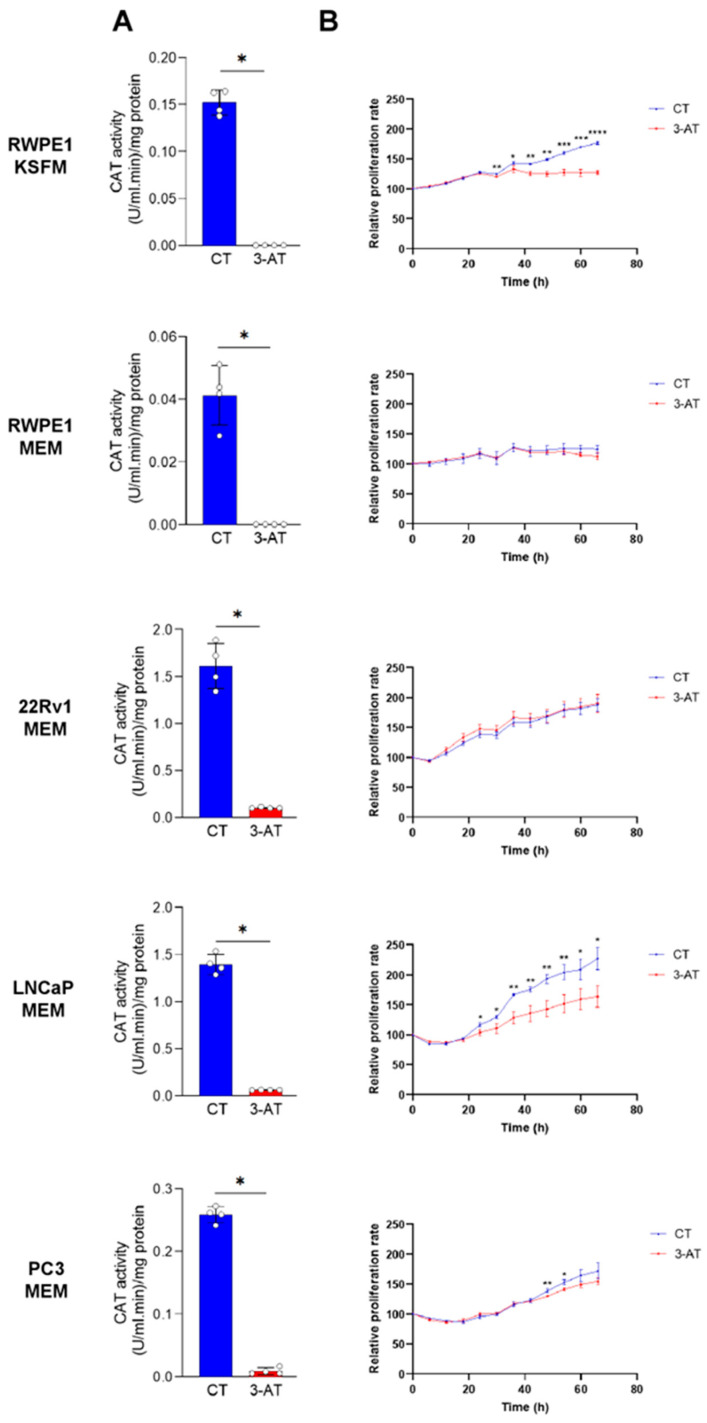
The catalase inhibitor 3-amino-1,2,4-triazole suppresses the growth of RWPE1/KSFM and LNCaP/MEM cells. (**A**) In all cell lines, 3-amino-1,2,4-triazole (3-AT) effectively inhibits CAT activity. Cells were treated with 10 mM 3-AT, and CAT activity was measured after two days in both control and 3-AT-treated conditions (n = 4). (**B**) Proliferation profiles of cells treated or not with 10 mM 3-AT. Data represent the mean relative confluence percentage compared to the starting point (n = 3). Error bars represent standard deviation. Statistical analysis was performed using the unpaired *T*-test (*, *p* < 0.05; **, *p* < 0.01; ***, *p* < 0.001; ****, *p* < 0.0001).

**Figure 8 antioxidants-13-01340-f008:**
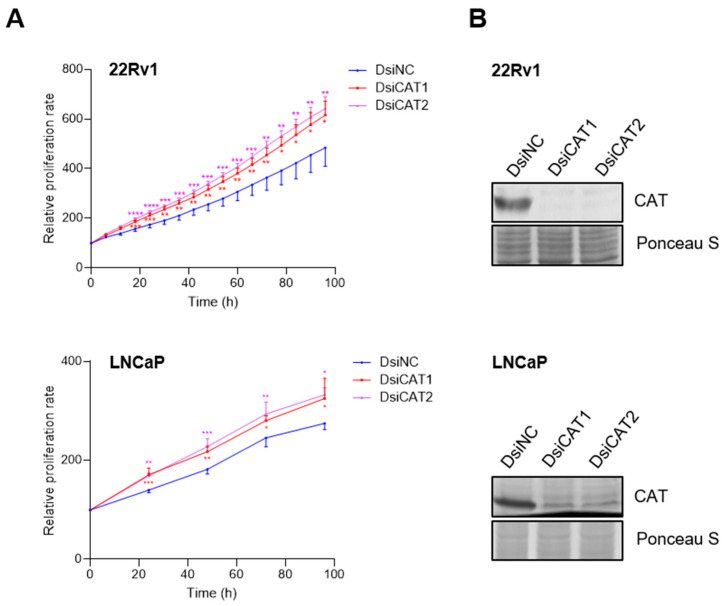
Downregulation of catalase expression promotes LNCaP and 22Rv1 cell proliferation. (**A**) Proliferation profiles of cells treated with DsiRNAs. Each dot represents the mean FC in confluence (relative to the starting point) from at least four independent biological replicates. Error bars indicate standard deviation. Statistical analysis was assessed at each time point using one-way ANOVA followed by a Dunnett’s test to compare DsiCAT1 and DsiCAT2 with DsiNC (*, *p* < 0.05; **, *p* < 0.01; ***, *p* < 0.001; ****, *p* < 0.0001). (**B**) Validation of the DsiCAT1 and DsiCAT2 siRNAs. Ponceau S staining was used to confirm equal protein loading.

**Figure 9 antioxidants-13-01340-f009:**
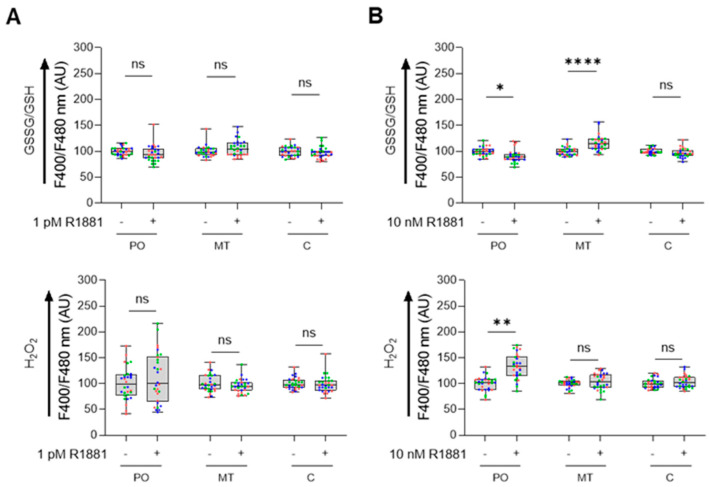
Treatment of LNCaP cells with 10 nM R1881 modulates the peroxisomal glutathione redox state and H_2_O_2_ levels. LNCaP cells were cultured in phenol red-free MEM supplemented with 10% charcoal-stripped serum for 4 days. Subsequently, they were treated with either the ethanol vehicle (−) or 1 pM or 10 nM R1881 (+). After one day, the cells were transfected with a plasmid encoding a peroxisomal (PO), cytosolic (C), or mitochondrial (MT) variant of (**A**) the glutathione redox sensor roGFP2, or (**B**) the H_2_O_2_ sensor roGFP2-Orp1 (representative images of the subcellular distribution patterns of the fluorescent reporter proteins are shown in Appendix A). Subsequently, the medium was replaced with fresh ethanol- or R1881-containing medium. Two days later, the F400/F480 response ratios of the sensors were measured, expressed as the percentage of the average vehicle response, and presented as box plots. Each box represents the interquartile range, with the bottom and top showing the 25th and 75th percentiles, respectively. The line inside the box indicates the median and the lines extending from the box representing one standard deviation below and above the mean. Data from each independent experiment (n = 3) are indicated by a different color. Statistical comparisons were performed using nested *T*-tests (ns, non-significant; *, *p* < 0.05; **, *p* < 0.01; ****, *p* < 0.0001). AU, arbitrary units.

**Figure 10 antioxidants-13-01340-f010:**
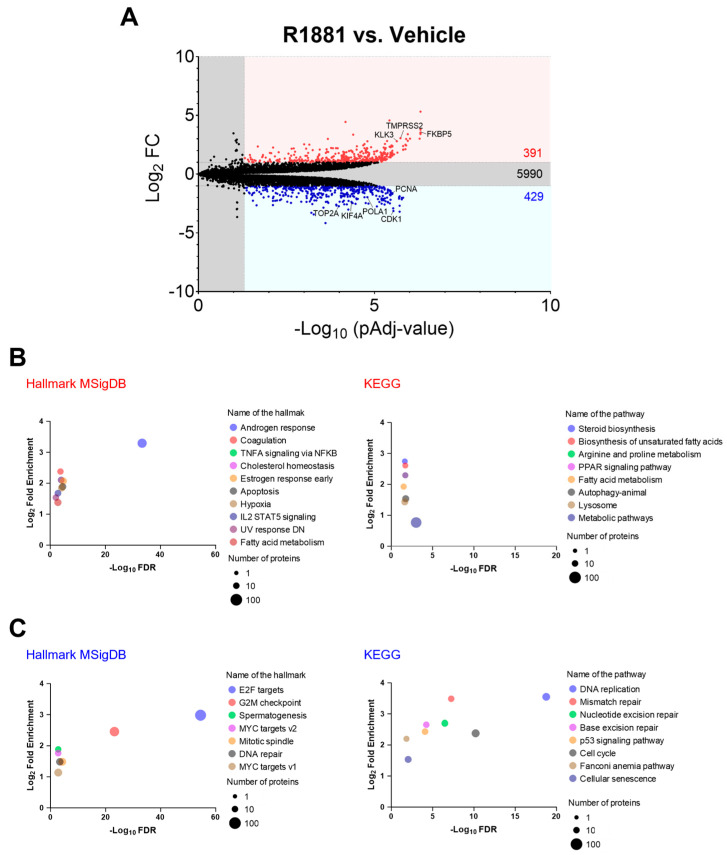
Gene-set enrichment analysis of differentially expressed proteins in LNCaP cells treated with the AR agonist R1881 compared to the vehicle. LNCaP cells were cultured in phenol red-free MEM supplemented with 10% charcoal-stripped serum for 4 days, followed by treatment with either ethanol (vehicle) or 10 nM R1881. After one day, the medium was replaced with fresh ethanol- or R1881-containing medium. Two days later, cells were harvested and processed for proteomics analysis. (**A**) Volcano plots showing the differences in protein abundance between 10 nM R1881-treated and vehicle-treated LNCaP cells. Proteins significantly up- or downregulated (FC ≥ 2; pAdj < 0.05) are indicated by red or blue dots, respectively (n = 3 biological replicates). Black dots represent proteins with no significant difference. Numbers within the panel denote the total number of proteins in each group. The list of differentially expressed proteins was generated using FragPipe-Analyst, and the Benjamin–Hochberg method was applied to correct for the FDR. (**B**,**C**) Molecular Signatures Database (MsigDB) hallmarks and KEGG enrichment analyses of the differentially expressed proteins were conducted using ShinyGO 0.80. Settings included selected species (human), FDR cut-off (0.05), and sorting by FDR. The total list of identified proteins was uploaded as the background. KEGG pathways and MsigDB hallmarks with significant enrichment (up to 10) are shown for the (**B**) upregulated and (**C**) downregulated differentially expressed proteins, sorted by fold enrichment. Circle size represents the number of proteins identified in each pathway/hallmark. DN, down-regulated; E2F, transcription factor E2F; IL2, interleukin 2; NFKB, nuclear factor kappa-B; PPAR, peroxisome proliferator-activated receptor; STAT5, signal transducer and activator of transcription 5; TNFA, tumor necrosis factor-alpha; UV, ultraviolet.

**Figure 11 antioxidants-13-01340-f011:**
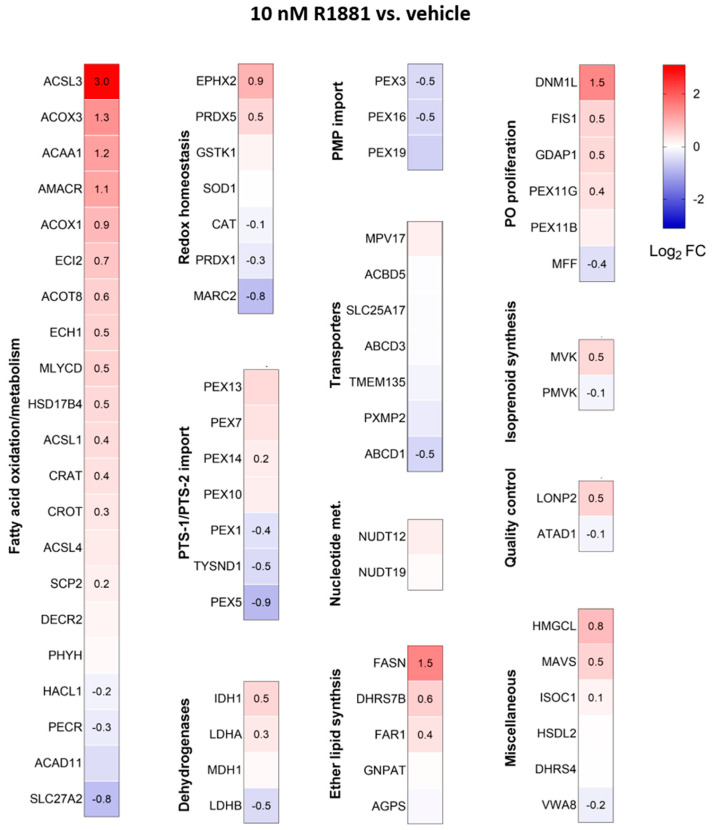
Heatmaps depicting the proteome profile of 72 peroxisome-associated proteins in LNCaP cells treated with the AR agonist R1881 relative to the vehicle. The relative protein abundances, expressed as Log_2_ FC, are shown on a false color scale with red for upregulation, white for equal expression, and blue for downregulation compared to vehicle-treated LNCaP cells. The cells were cultured in phenol red-free MEM supplemented with 10% (*v*/*v*) charcoal-stripped serum for 4 days, followed by treatment with either ethanol (vehicle) or 10 nM R1881. After one day, the medium was replaced with fresh ethanol- or R1881-containing medium. Two days later, cells were harvested and processed for proteomics analysis. Each box represents the average of 3 biological replicates, with Log_2_-FC shown only if the adjusted *p*-value is <0.05.

**Figure 12 antioxidants-13-01340-f012:**
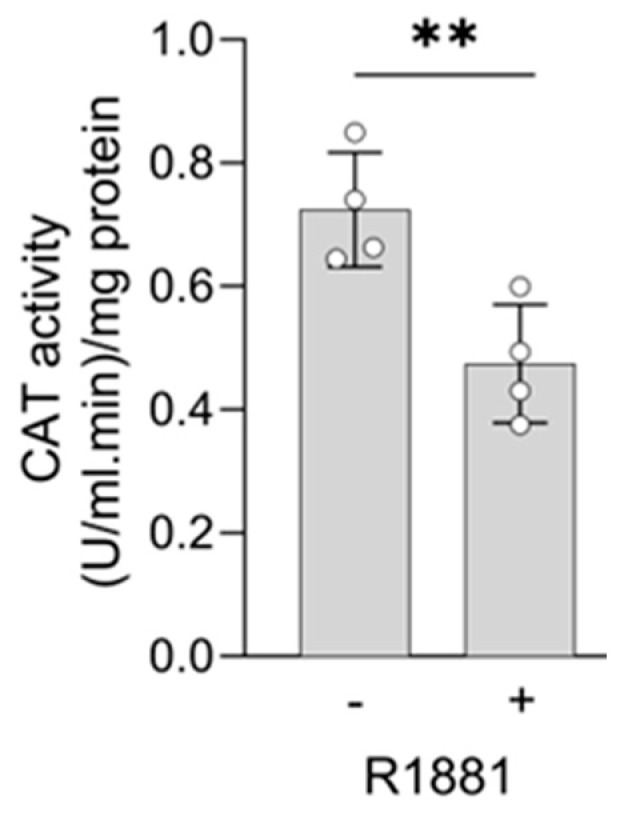
The AR agonist R1881 decreases catalase activity in LNCaP cells. LNCaP cells were cultured in phenol red-free MEM supplemented with 10% (*v*/*v*) charcoal-stripped serum for 4 days. The cells were then treated with either ethanol (vehicle) or 10 nM R1881. After one day, the medium was replaced with fresh medium containing ethanol, or R1881. Two days later, CAT activity was measured (n = 4). Error bars represent the standard deviation. Statistical significance was assessed using an unpaired *T*-test (**, *p* < 0.01).

**Figure 13 antioxidants-13-01340-f013:**
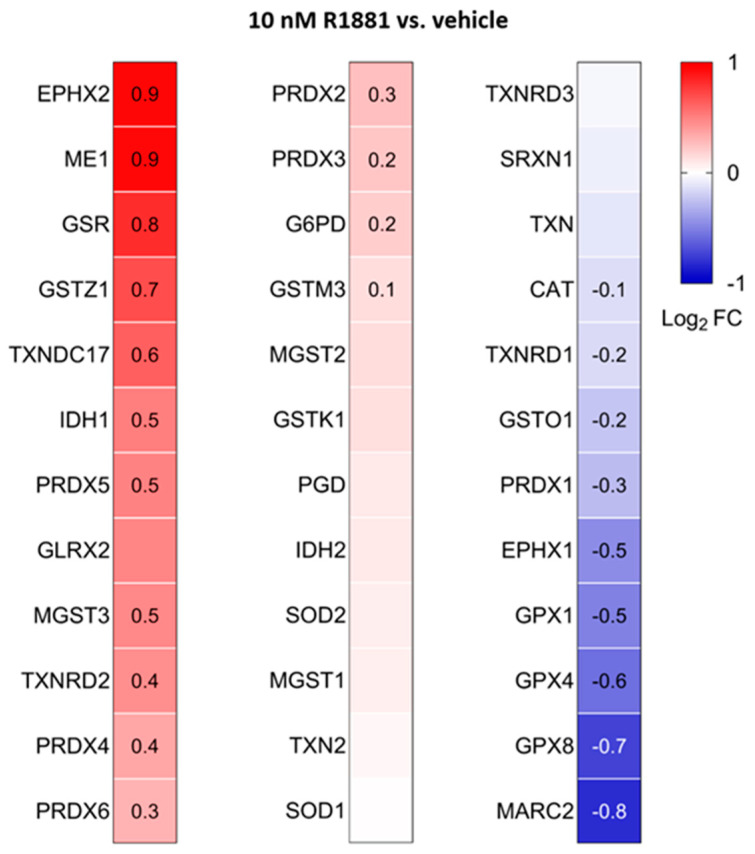
Heatmaps showing the proteome profile of 36 primary and secondary antioxidant enzymes in LNCaP cells treated with the AR agonist R1881 compared to the vehicle. The cells were cultured in phenol red-free MEM supplemented with 10% (*v*/*v*) charcoal-stripped serum for 4 days. The cells were then treated with either ethanol (vehicle) or 10 mM R1881. After one day, the medium was replaced with fresh medium containing either ethanol or R1881. Two days later, cells were harvested and processed for proteomics analysis. Each box represents the average of 3 biological replicates, with Log_2_ FC shown only if the adjusted *p*-value is <0.05.

## Data Availability

The raw data from the LC-MS runs can be accessed via ProteomeXchange (PRIDE database) using the following identifiers: (1) PXD054270 (name: peroxisomal proteome of prostate (cancer) cells); and (2) PXD054374 (name: R1881 treatment modulates LNCaP proteome).

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
