# Peer review of "Characterization of the Peroxisomal Proteome and Redox Balance in Human Prostate Cancer Cell Lines"

_antioxidants, 2024, doi:10.3390/antiox13111340_

Round 1
Reviewer 1 Report
In the current paper, the authors performed comprehensive proteomic analyses to compare peroxisomal and redox protein profiles between benign (RWPE-1) and malignant (22Rv1, LNCaP, and PC3) prostate cell lines. Our analyses revealed a significant enrichment of the "peroxisome" pathway among proteins significantly upregulated in androgen receptor (AR)-positive cell lines. The results are potentially interesting. However, the conclusion is not sufficiently supported by the authors' results. The specific points are as follows.
Major points.
1. The authors claim that increased AR activity reduces CAT function, leading to increased peroxisomal H2O2 levels that trigger adaptive stress responses to promote cell survival, growth and proliferation in prostate cell lines. However, there is no correlation between individual catalase activity and AR activity in prostate cell lines with high (22Rv1), intermediate (LNCaP) and low (PC3) catalase activity. Therefore, it is not sufficient to draw this conclusion. The authors show that treatment of LNCaP with the AR agonist R1881 decreased catalase activity and altered gene expression, but it is not surprising that treatment with the drug caused some changes. The problem is that they only used one cell type, LNCaP, and the relationship between catalase activity and AR activity is unclear.
2. The heatmap shows that peroxisome-associated proteins are increased by the heatmap, but what is the conclusion about the relationship between each catalase activity in in prostate cell lines with high, medium, and low catalase activity? The relative protein abundances seem to all increase in the same way; are there any statistically significant differences in the expression changes in the three prostate cell lines? If not, then this does not reflect a role for catalase in them.
Minor points.
1. Figure 6B: A loading control such as beta-actin should be included.
2. English should be carefully edited by a professional English editing service.
3. Catalase activity of 22Rv cells was significantly inhibited by 3-AT treatment, but cell proliferation was unchanged (Fig. 7). However, knockdown of catalase significantly reduced cell proliferation (Fig. 8). A rational explanation is required.
Author Response
We value the reviewer’s feedback on our work and trust that the following responses will adequately address the points raised. Note that the figure, line, and reference numbers correspond to those in the revised manuscript.
The title of the manuscript is like that of a review article. It needs to be changed to a more appropriate title
As requested by the reviewer, we have revised the original title, “Peroxisomes and Redox Balance in Human Prostate Cancer Cell Lines: The Role of Catalase”, to “Characterization of the Peroxisomal Proteome and Redox Balance in Human Prostate Cancer Cell Lines.” (see lines 2-3). We hope that the inclusion of the term “characterization” clarifies that this is an experimental study, not a review article.
Is it necessary to include study limitations in the discussion? Yes
In response to the request, we have added a new paragraph to the Discussion that addresses the study’s limitations (see lines 701-709). The revised text now reads: “Lastly, we acknowledge several limitations of our study. First, our proteomics analyses did not identify some well-known peroxisomal proteins (e.g., D-amino acid oxidase, 2-hydroxyacid oxidase, polyamine oxidase, pipecolic acid oxidase, and sarcosine oxidase). In addition, while we observed changes in the cytosolic glutathione redox state (Figure 1) and the protein levels of enzymes involved in NADPH-dependent glutathione (GSH) regeneration and the maintenance of the NADP+/NADPH balance (Figure 5), we did not directly measure the levels of these pyridine nucleotides. Furthermore, since our findings are primarily based on commonly used prostate cell lines, it is important to validate these results in patient tissue samples.” We hope this revision adequately addresses the reviewer’s concerns.
Major points.
- The authors claim that increased AR activity reduces CAT function, leading to increased peroxisomal H2O2 levels that trigger adaptive stress responses to promote cell survival, growth and proliferation in prostate cell lines. However, there is no correlation between individual catalase activity and AR activity in prostate cell lines with high (22Rv1), intermediate (LNCaP) and low (PC3) catalase activity. Therefore, it is not sufficient to draw this conclusion. The authors show that treatment of LNCaP with the AR agonist R1881 decreased catalase activity and altered gene expression, but it is not surprising that treatment with the drug caused some changes. The problem is that they only used one cell type, LNCaP, and the relationship between catalase activity and AR activity is unclear.
In response to the reviewer’s comment about the lack of correlation between CAT and AR activity, we would like to clarify that establishing such correlations among different cell types at basal levels may not be appropriate for several reasons. First, it is important to recognize that catalase expression can be regulated by various transcription factors, as well as through genetic, epigenetic, and posttranscriptional processes [51]. One example is the androgen receptor-mediated suppression of catalase via FOXO3A-dependent signaling [52] (see lines 632-635). Second, both 22Rv1 and LNCaP cells are AR-positive, but only LNCaP cells rely on this transcription factor for their growth [DOI: 10.1074/jbc.273.32.20213; DOI: 10.1007/s11626-999-0115-4]. In addition, PC3 cells are AR-negative, making it inappropriate to compare the effects of AR activation on CAT activity and cell growth across these cell lines. To enhance clarity for readers, we have specified that our conclusion regarding the observed correlation between AR and CAT applies only to AR-positive cell lines (see lines 614-615).
In response to the reviewer’s comment regarding the use of LNCaP cells and the changes observed following treatment with the AR agonist R1881, we would like to clarify that we chose LNCaP cells due to their androgen dependency for growth [DOI: 10.1074/jbc.273.32.20213]. However, as shown in Figure S13, we also observed a decrease in catalase expression in 22Rv1 cells following treatment with R1881. In addition, as mentioned above, a recent study [52] reported that AR activity regulates catalase expression in LNCaP cells via Foxo3A transcription. These findings are further supported by recent mRNA expression data from 22Rv1, VCaP, and LNCaP cells available in the Gene Expression Omnibus (GEO) database, with accession numbers GSE130534 [47], GSE220618 [66], GSE247592 [67], and GSE214756 [68]. This information has now been included in the manuscript (see lines 689-692 and Figure S10).
- The heatmap shows that peroxisome-associated proteins are increased by the heatmap, but what is the conclusion about the relationship between each catalase activity in prostate cell lines with high, medium, and low catalase activity? The relative protein abundances seem to all increase in the same way; are there any statistically significant differences in the expression changes in the three prostate cell lines? If not, then this does not reflect a role for catalase in them.
To address the concern regarding the relationship between catalase activity and protein expression in the prostate cancer cell lines, we would like to clarify the following points based on our data. As shown in the heatmaps (Figure 4), several peroxisome-associated proteins exhibit altered relative abundances, with some increasing (e.g., ACAA1) and others decreasing (e.g., ACAD11) in prostate cancer (PCa) cell lines compared to the RWPE-1 cell line. However, when comparing the relative abundances of peroxisome-associated proteins among the different PCa cell lines, we also observe statistically significant differences. For the reviewer’s convenience, we have included a figure showing volcano plots of CAT and all identified proteins involved in peroxisomal α- or β-oxidation (see Figure R1, for review purposes only). As illustrated in this figure, while the protein levels of catalase remain statistically unchanged, the H2O2-producing enzymes ACOX1 and/or ACOX3 show significant alterations.
Minor points.
- Figure 6B: A loading control such as beta-actin should be included.
An updated blot has been provided, using GAPDH as the loading control.
- English should be carefully edited by a professional English editing service.
We value the reviewer’s feedback regarding the quality of the English in our manuscript, despite his/her expressed uncertainty about his/her qualifications to evaluate it. In this context, it is important to note that the other reviewer found the English language to be adequate. Nevertheless, in response to this criticism, we enlisted the help of Dr. Paul Walton (University of Western Ontario, Canada), a native English-speaking colleague, to enhance the manuscript’s language. He has confirmed that the current version is clear and fully understandable for both native English speakers and those for whom English is a second language. We acknowledge his contribution in lines 766-768 of the acknowledgment section.
- Catalase activity of 22Rv cells was significantly inhibited by 3-AT treatment, but cell proliferation was unchanged (Fig. 7). However, knockdown of catalase significantly reduced cell proliferation (Fig. 8). A rational explanation is required.
As noted in lines 663-665, previous studies have demonstrated that 3-AT, despite its widespread use as a catalase inhibitor, has several off-target effects. Specifically, this compound can inhibit α-oxidation of fatty acids [58], heme synthesis [59], and the activity of cytochrome P450 2E1 [60]. Consequently, we aimed to confirm our findings regarding 3-AT by using two different DsiRNAs to downregulate catalase expression. Notably, while treatment with 3-AT had little to no inhibitory effect on the proliferation of PC3 and 22Rv1 cells (Figure 7B), DsiRNA-mediated knockdown of catalase resulted in increased proliferation of these cells (Figure 8). Given that reduced CAT activity can lead to elevated physiological levels of H2O2, which may promote signaling pathways that drive cancer cell metabolism and proliferation [19], our findings are not entirely surprising. In addition, it is important to consider the findings of another study [57], which reported that a concentration of 250 mM 3-AT inhibited PC3 cell proliferation; however, this result should be interpreted with caution due to the off-target effects associated with 3-AT. All this information has been included in the revised manuscript (see lines 653-665).
Reviewer 2 Report
The manuscript submitted by Mohamed A. F. Hussein and colleagues investigates the significance of the peroxisome pathway in prostate cancer cell lines. The study highlights the upregulation of this pathway, along with catalase levels, in androgen receptor-positive prostate cancer cells. The experimental work, particularly the proteomics analysis, is well-executed and offers valuable insights. However, before proceeding to publication, I recommend addressing the following concerns:
- The primary weakness of the study is the inconsistency between pharmacological inhibition of catalase and its knockdown using siRNA. While you mention the potential non-specificity of 3-AT, it is unexpected that catalase inhibition leads to increased proliferation in cells with the highest enzyme levels. Did you assess cell viability in catalase-inhibited cells in parallel with proliferation? This data could offer insights into the observed proliferation effects.
- AR levels and androgen dependency: Did you monitor androgen receptor (AR) levels, localization, or androgen-dependency in AR-positive cells following catalase inhibition? This may help clarify the unexpected increase in proliferation upon catalase inhibition.
- Can you provide data on the efficiency of transfection using lipofectamine? Did you use any method for selecting transfected cells, such as puromycin resistance?
- - Additional experiments with antiandrogens: In addition to your experiments using R1881, I suggest conducting similar experiments (or at least measuring catalase activity) with antiandrogens such as bicalutamide or flutamide, or using charcoal-stripped media to explore androgen-dependency further.
- - Considering the absence of in vivo data, I recommend including a supplementary analysis of catalase expression in different Gleason scores or prostate cancer subtypes using data from The Cancer Genome Atlas (TCGA).
-
Figure 6B: A loading control such as Ponceau staining or a housekeeping protein is missing in Figure 6B.
Figure 7, Please adjust the scales of the CAT activity graphs for each cell line individually, as using the same scale makes it difficult to discern differences in certain cases.
Author Response
We are grateful for the reviewer’s insights into our work and trust that the following responses will thoroughly address the issues highlighted. Note that the figure, line, and reference numbers correspond to those in the revised manuscript.
Is it necessary to include study limitations in the discussion? Yes
Please see our responses to the same issue raised by Reviewer 1.
The primary weakness of the study is the inconsistency between pharmacological inhibition of catalase and its knockdown using siRNA. While you mention the potential non-specificity of 3-AT, it is unexpected that catalase inhibition leads to increased proliferation in cells with the highest enzyme levels. Did you assess cell viability in catalase-inhibited cells in parallel with proliferation? This data could offer insights into the observed proliferation effects.
In response to the reviewer’s concern regarding the inconsistency between pharmacological inhibition of catalase and its knockdown using siRNA, we would like to clarify the following. It is expected that in cells with high basal catalase levels, knocking down this antioxidant enzyme would lead to a more pronounced increase in H2O2 levels compared to cells with lower catalase expression. As elevated physiological H2O2 levels are known to stimulate cell proliferation [19], our findings are consistent with this mechanism (see also Reviewer 1, minor point 3).
To address the reviewer’s inquiry, we conducted a sulforhodamine B assay, a widely used and reliable method for indirectly assessing cell growth, proliferation, and viability. The results from this assay confirmed our IncuCyte data (see lines 477-478 and Figure S5).
AR levels and androgen dependency: Did you monitor androgen receptor (AR) levels, localization, or androgen-dependency in AR-positive cells following catalase inhibition? This may help clarify the unexpected increase in proliferation upon catalase inhibition.
Thank you for your suggestion. In response, we conducted additional experiments to address this point. Our results show that reducing catalase levels did not impact AR expression or activity, as evidenced by prostate-specific antigen staining (see lines 588-591 and Figure S11). These findings suggest that CAT likely functions downstream, rather than upstream, of the AR signaling pathway. However, as AR already displays predominantly nuclear localization in LNCaP cells upon culturing in charcoal-stripped 10% FBS (Biowest, S181F-100; this serum still contains 0.2 nM testosterone) (see Figure R2, for review purposes only), our immunofluorescence data did not provide sufficient evidence to assess the effect of CAT depletion on AR’s subcellular localization. Given the need for more in-depth analysis and the limited time available for revisions, we prefer not to draw definitive conclusions from these experiments at this stage but plan to explore this further in future studies. We hope the reviewer agrees with this approach.
Can you provide data on the efficiency of transfection using lipofectamine? Did you use any method for selecting transfected cells, such as puromycin resistance?
Transfection efficiency was evaluated using a fluorescent TYE 563-labeled Transfection Control DsiRNA (IDT, 51-01-20-19) and found to be nearly 100% (see Figure R3, for review purposes only). This information has also been added to the Materials and Methods section, lines 286-287). Note that this observation aligns with our immunoblotting results, which show that residual catalase levels are near or below detection limits (see Figure 8). Because we used DsiRNAs, not plasmids encoding shRNAs, no selection method, such as puromycin resistance, was applied.
Additional experiments with antiandrogens: In addition to your experiments using R1881, I suggest conducting similar experiments (or at least measuring catalase activity) with antiandrogens such as bicalutamide or flutamide, or using charcoal-stripped media to explore androgen-dependency further.
As suggested by the reviewer, we have now included an experiment in which LNCaP cells were treated with an anti-androgen. Instead of using bicalutamide or flutamide, we opted for enzalutamide, a second-generation anti-androgen. Our results indicate that this treatment did not significantly affect catalase activity or subcellular H2O2 levels (Figure S9). Note that this lack of phenotypic change is consistent with transcriptome data indicating that enzalutamide treatment results in only a slight, non-significant increase in CAT levels (Figure S10C) [47], likely due to the already low androgen levels in the culture medium [48] (see lines 584-588).
Considering the absence of in vivo data, I recommend including a supplementary analysis of catalase expression in different Gleason scores or prostate cancer subtypes using data from The Cancer Genome Atlas (TCGA).
In response to the reviewer’s suggestion, we have added a figure that illustrates catalase expression across various types of prostate cancer tissues (see line 643 and Figure S12). This figure was created using expression data obtained from the University of Alabama at Birmingham Cancer Data Analysis Portal.
Figure 6B: A loading control such as Ponceau staining or a housekeeping protein is missing in Figure 6B.
An updated blot has been provided, using GAPDH as the loading control.
Figure 7, Please adjust the scales of the CAT activity graphs for each cell line individually, as using the same scale makes it difficult to discern differences in certain cases.
As requested by the reviewer, we have adjusted the Y-axes for the CAT activity graphs for each cell line separately.
Round 2
Reviewer 1 Report
The authors have addressed almost all of my concerns.
The authors have addressed almost all of my concerns.
Reviewer 2 Report
I thank the authors for addressing all the concerns.
-